# Possibility of Amino Acid Treatment to Prevent the Psychiatric Disorders via Modulation of the Production of Tryptophan Metabolite Kynurenic Acid

**DOI:** 10.3390/nu12051403

**Published:** 2020-05-13

**Authors:** Tsutomu Fukuwatari

**Affiliations:** Department of Nutrition, School of Human Cultures, the University of Shiga Prefecture, 2500 Hassaka, Hikone, Shiga 522-8533, Japan; fukkie@shc.usp.ac.jp; Tel.: +81-749-28-8443

**Keywords:** dopamine, kynurenic acid, kynurenine, large neutral amino acid transporter, neuropsychiatric disorders, neurotransmitter, α7 nicotinic acetylcholine receptor, *N*-methyl-d-aspartic acid (NMDA) receptor, tryptophan

## Abstract

Kynurenic acid, a metabolite of the kynurenine pathway of tryptophan catabolism, acts as an antagonist for both the α7 nicotinic acetylcholine receptor and glycine coagonist sites of the *N*-methyl-d-aspartic acid receptor at endogenous brain concentrations. Elevation of brain kynurenic acid levels reduces the release of neurotransmitters such as dopamine and glutamate, and kynurenic acid is considered to be involved in psychiatric disorders such as schizophrenia and depression. Thus, the control of kynurenine pathway, especially kynurenic acid production, in the brain is an important target for the improvement of brain function or the effective treatment of brain disorders. Astrocytes uptake kynurenine, the immediate precursor of kynurenic acid, via large neutral amino acid transporters, and metabolize kynurenine to kynurenic acid by kynurenine aminotransferases. The former transport both branched-chain and aromatic amino acids, and the latter have substrate specificity for amino acids and their metabolites. Recent studies have suggested the possibility that amino acids may suppress kynurenic acid production via the blockade of kynurenine transport or via kynurenic acid synthesis reactions. This approach may be useful in the treatment and prevention of neurological and psychiatric diseases associated with elevated kynurenic acid levels.

## 1. Introduction

The essential amino acid tryptophan is well known as a precursor of several bioactive compounds such as serotonin and melatonin. More than 90% of tryptophan is metabolized by the kynurenine pathway [1], and this pathway plays a critical role in tryptophan catabolism and coenzyme nicotinamide adenine dinucleotide (NAD^+^) supply (Figure 1). Recently, many researchers have studied the kynurenine pathway, because the pathway has interesting intermediates and metabolites. For example, kynurenine regulates immunoreaction as an aryl hydrocarbon receptor agonist [2], and kynurenic acid (KYNA) affects brain function as an antagonist for both the α7 nicotinic acetylcholine receptors (α7nAchRs) and the *N*-methyl-d-aspartic acid (NMDA) receptor [3,4] and an agonist for the G protein-coupled receptor (GPR) 35 (GPR35) [5]. 3-hydroxykynurenine is a potential endogenous neurotoxin and oxidative stress generator [6], and quinolinic acid produces excitotoxicity as an NMDA receptor agonist [7]. Especially, KYNA function research has dramatically developed since 2001, and one of the targets for KYNA research is to manipulate KYNA production in the brain to prevent and improve psychiatric disorders such as schizophrenia and depression. In the present article, we briefly review recent advances in KYNA research and further describe the ability of amino acids to modulate KYNA production. The structure of tryptophan, kynurenine, and KYNA are shown in Figure 2.

## 2. Function of Kynurenic Acid in the Brain

In 1989, Kessler et al. found that KYNA competitively inhibited glycine coagonist site of the NMDA receptor at low concentration with an IC_50_ of 8 μmol/L [3]. A decade later, Hilmas et al. found that KYNA noncompetitively inhibited α7nAchRs with an IC_50_ of 7 μmol/L using the patch-clamp technique with cultured hippocampal neurons [4]. Furthermore, Wang et al. found that KYNA is ligand for GPR35, whose EC_50s_ are 10.7, 7.4, and 39.2 μmol/L in mouse, rat, and human, respectively [5]. Since physiological concentrations of brain KYNA are 5 pmol/g wet wt, 15 pmol/g wet wt, and 150 pmol/g wet wt in mouse, rat and human, respectively [8], elevation of brain KYNA has been considered to affect these receptors. Effects of KYNA increase on the neurotransmitter release were investigated using microdialysis technique, and KYNA concentration-dependently and reversibly reduced extracellular glutamate, dopamine, and γ-aminobutyric acid (GABA) to less than 50% of baseline concentrations [9,10,11]. Conversely, inhibition of endogenous KYNA formation by reverse dialysis of KYNA synthesis inhibitor *(S)*-4-(ethylsulfonyl) benzoylalanine (*S*-ESBA) reversibly increases dopamine, glutamate, and GABA levels in the rodent brain [11,12,13]. Although these findings suggest that changes of brain KYNA levels affect neurotransmitter release via modulation of above receptors, understanding the mechanism of action of KYNA is difficult. There is disagreement about the interaction between KYNA and α7nAchRs, because several studies failed to reproduce evidence for action of KYNA on nicotinic receptors. [14]. Schematic representation of the interaction between KYNA and the receptors is shown in Figure 3.

Behavioral studies showed that changes in brain KYNA levels affect several psychiatric functions in experimental animals. Elevation of endogenous KYNA concentration produces disruptions in prepulse inhibition [15] and habituation of auditory-evoked potentials [16], indicating that elevated KYNA levels interfere with normal reductions in processing and responding to irrelevant stimuli. Elevation of brain KYNA levels also affects cognitive function. For example, rats with elevations of endogenous KYNA exhibit spatial working memory deficits in a radial arm maze task [17]. These rats also exhibit impaired contextual fear memory consisting of two pairings of a tone and foot shock, and are slower to learn to discriminate between different contexts with or without foot shock [18]. On the other hand, reduction of endogenous KYNA levels by genetic and pharmacological manipulation improves cognitive functions. Mice with a targeted deletion of kynurenine aminotransferase II (KAT II), a major biosynthetic enzyme of brain KYNA, show reduced brain KYNA levels and significantly increased performance in three cognitive paradigms that rely in part on the integrity of hippocampal function, namely, object exploration and recognition, passive avoidance, and spatial discrimination [19]. Intracerebroventricular administration of selective KAT II inhibitor *S*-ESBA improves kynurenine-induced cognitive deficits on performance in the Morris water maze [20]. Systemic administration of KAT II inhibitor, PF-04859989, also dose-dependently reduces brain KYNA, prevents amphetamine- and ketamine-induced disruption of auditory gating, and improves performance in a sustained attention task [21]. It also prevents ketamine-induced disruption of performance in a working memory task and a spatial memory task in rodents and nonhuman primates, respectively. These findings support the hypotheses that endogenous KYNA impacts cognitive function and that inhibition of KAT II, and consequent lowering of endogenous brain KYNA levels, improves cognitive performance under conditions considered relevant for schizophrenia.

In humans, elevated KYNA levels are observed in the cerebrospinal fluid and cortex of patients with schizophrenia and bipolar disorder [22,23,24,25,26,27]. In the brain, kynurenine and KYNA levels in schizophrenic cases are 1.5 times higher than matched control subjects [24]. Similar observations reported that kynurenine and KYNA concentrations in the cerebrospinal fluid (CSF) were 2 and 1.5 times higher in patients with schizophrenia, respectively, than with healthy volunteers, whereas tryptophan concentrations did not differ between the groups [26]. Patients with bipolar disorder have 1.5 times increased levels of KYNA in their CSF compared with healthy volunteers, and the levels of KYNA are positively correlated with age among bipolar patients but not in healthy volunteers [28]. Haplotype analysis shows an association between kynurenine 3-monoxygenase (KMO) gene polymorphisms and CSF concentrations of KYNA in patients with schizophrenia [29]. In the bipolar disorder and schizophrenia patients, KMO mRNA levels are reduced in the brain compared with nonpsychotic patients and controls, and the KMO Arg452 allele is associated with increased levels of CSF KYNA and reduced brain KMO expression [30]. KMO is the primary enzyme responsible for kynurenine degradation. These results support the hypothesis that KYNA is involved in the pathophysiology of psychiatric diseases such as schizophrenia and bipolar disorder.

## 3. Kynurenic Acid Synthesis

Tryptophan degradation is initiated by tryptophan 2,3-dioxygenase (TDO) and indoleamine 2,3-dioxygenase (IDO), and these enzymes metabolize tryptophan to *N*-formylkynurenine, which is further degraded to kynurenine by formamidase. Kynurenine is catabolized to KYNA, 3-hydroxykynurenine, and anthranilic acid by KAT, KMO, and kynureninase, respectively. In the brain, 3-hyroxykynurenine and further downstream kynurenine pathway metabolites are synthesized in microglia, whereas KYNA is formed in astrocytes [31]. Approximately 40% of the kynurenine in brain is synthesized in astrocytes from tryptophan, and the remainder comes from plasma [32]. TDO-deficient mice show higher plasma tryptophan and kynurenine levels [33], and IDO-deficient mice show normal level of serum tryptophan and very low level of kynurenine [34]. These results suggest that most tryptophan is degraded by TDO in the liver, and that plasma kynurenine is derived from nonhepatic tissues and produced by IDO rather than TDO. Skeletal muscle also affects plasma kynurenine levels. Exercise training increases murine and human KAT expression in the skeletal muscle, and decreases plasma kynurenine levels by enhancement of kynurenine expenditure in the skeletal muscle [35]. Figure 4 shows the organ–organ interactions for tryptophan and kynurenine metabolism (Figure 4).

Astrocytes uptake peripheral kynurenine from the blood stream via large neutral amino acid transporters (LATs). There is poor transport of KYNA across the blood–brain barrier, and for this reason plasma KYNA is not expected to contribute significantly to the brain KYNA pool [36]. LATs are known to transport both branched chain amino acids (e.g., valine, leucine, and isoleucine) and aromatic amino acids (e.g., tyrosine, phenylalanine, and tryptophan). Several findings show that LATs transport amino acids with higher affinity than kynurenine in tumor cell lines [36,37,38,39]. There are two LATs (LAT 1 and LAT 2); the affinity of LAT 1 to large neutral amino acids is higher than that of LAT 2. LAT 1 exhibits high-affinity transport of large neutral amino acids, including branched chain and aromatic amino acids, while LAT 2 transports not only large neutral amino acids but also small neutral amino acids in a fashion that appears to have broader substrate selectivity than LAT 1. [40,41]. LAT 1 is expressed in brain, spleen, placenta, testis, colon, and tumor cells, whereas LAT 2 is expressed at high levels in the small intestine, kidney, brain, and skeletal muscle. Neither of the LATs is expressed in the liver. The *K*_m_ value of LATs for kynurenine is ~160 µmol/L, 80 times higher than plasma kynurenine concentrations [36,37], whereas the *K*_m_ values of LAT 1 for leucine, isoleucine, methionine, phenylalanine, tyrosine, and histidine are 15–30 µmol/L, which are at physiological concentrations [39].

KATs catalyze the irreversible transamination reaction of kynurenine to KYNA. Four KATs have been identified in the mammalian brain: KAT I (glutamine transaminase K, GTK; EC 2.6.1.64), KAT II (2-aminoadipate aminotransferase, ADA; EC 2.6.1.7), KAT III (cysteine conjugate β-lyase 2, CCBL2; EC4.4.1.13), and KAT IV (mitochondrial aspartate aminotransferase, ASAT; EC 2.6.1.1). The *K*_m_ values of KAT I, II, III, and IV for kynurenine are 875 μmol/L, 660 μmol/L, 1.5 mmol/L and 724 μmol/L, respectively [42,43]. Specificity for substrates is different among KATs. KAT I is inhibited by glutamine (IC_50_: 0.2 mmol/L); KAT II by lysine metabolite 2-aminoadipic acid, quisqualate, aspartate, and glutamate (IC_50_: 0.006, 0.02, 1.2 and 2.1 mmol/L, respectively); KAT III by methionine, glutamine, histidine, and cysteine; and KAT IV by quisqualate, aspartate, glutamate, and 2-aminoadipic acid (IC_50_: 0.1, 0.3, 0.9 and 1.5 mmol/L, respectively) [42,43]. KAT II activity accounts for highest proportion (60%) of the total KAT activity in the rats and human brain, with 10 and 30% contributed by KAT I and IV, respectively. In mice brain, KAT IV is the dominant KAT with 60% of total KAT activity [42]. KAT III contribution to brain KYNA synthesis remains to be determined. These findings suggest that KAT II plays a central role for KYNA synthesis in the brain, and thus KAT II can be targeted to regulate KYNA production. KAT II-deficient mice exhibit lower KYNA levels in the brain and increased performance in cognitive functions [19,44]. KAT II inhibitors successfully prevent elevation of brain KYNA levels and cognitive dysfunction [20,21].

Several factors increase KYNA production in the brain in vivo. Systemic administration of kynurenine and KMO inhibitor increase brain KYNA levels by elevation of blood kynurenine levels [45,46]. Chronic exposure to a high-fat and low-protein/carbohydrate ketogenic diet shows a several-fold increase in KYNA concentrations in the rat striatum and hippocampus [47]. Experimental diabetes mellitus type 1 enhances KAT II activity and increases KYNA levels in the rat cortex and hippocampus [48]. Thioacetamide-induced acute liver failure enhances peripheral kynurenine production, and thus increases brain KYNA levels [49]. Acute stress increases brain KYNA levels in the fetus and adulthood [50,51,52], and reduction of stress-increased KYNA prevents the impairment of fear discrimination [52]. KMO gene polymorphisms influence CSF KYNA levels in patients with schizophrenia and bipolar disorder [29,30].

As the de novo synthesized KYNA immediately liberates to the extracellular compartment, extracellular KYNA levels are dependent on KYNA production, which is regulated by two key factors: KAT activity and the availability of the KYNA substrate kynurenine [53]. Kynurenine is produced in the peripheral tissues, and astrocytes uptake kynurenine from blood stream via LATs, catalyze kynurenine to KYNA via KATs, and then excrete KYNA to the extracellular compartment. Therefore, four points can affect brain KYNA levels: (1) kynurenine formation in the peripheral tissues, (2) kynurenine uptake by astrocytes, (3) KYNA synthesis by KATs, and (4) KYNA release from astrocytes. Enhancement of KAT II activity in the skeletal muscle by exercise is an example to modulate peripheral kynurenine formation [35]. Although organic anion transporters 1 and 3 transport KYNA, and both transporters express in the brain and kidney [54], little information is available for KYNA release from astrocytes. KYNA is also released from other KYNA-producing tissues, including the liver and skeletal muscle to blood stream, and then excreted to urine. The liver and skeletal muscle dominantly metabolize kynurenine to KYNA, and small amount of brain-derived KYNA contributes to plasma KYNA levels. Since elevated inflammatory activity may drive elevations of kynurenine and KYNA levels through the activation of IDO, peripheral KYNA measurements have been expected to be a predictor of central KYNA levels. However, studies of peripheral KYNA concentrations in psychiatric disorders have reported conflicting results [55,56,57]. Although enhancement of peripheral kynurenine production increases both peripheral and brain KYNA levels, that of brain kynurenine uptake and KATs activities does not always affect peripheral KYNA production.

## 4. Effects of Amino Acids on Kynurenic Acid Production

High tryptophan diets increase brain KYNA levels owing to increased peripheral kynurenine in a dose-dependent manner, and reduce dopamine release via enhancement of KYNA production in the rat striatum [58]. The plausible mechanism to increase brain KYNA levels is that the peripheral tissues produce more kynurenine from the high dose of tryptophan and release more kynurenine into the blood stream, and astrocytes take up the more circulating kynurenine and metabolize more kynurenine to KYNA by KATs. Chronic intake of 5 g/d of tryptophan shows two-fold of increase serum kynurenine concentration in healthy volunteers, suggesting increase of brain KYNA production due to excess tryptophan intake in humans [59].

As mentioned above, to modulate KYNA production in the brain, four points are relevant: (1) kynurenine formation in the peripheral tissues, (2) kynurenine uptake by astrocytes, (3) KYNA synthesis by KATs, and (4) KYNA release from astrocytes. Since LATs transport kynurenine and amino acids, including both branched chain and aromatic amino acids, and KATs have broad substrate specificity, including amino acids and its metabolites, amino acids have the potential to suppress KYNA production via inhibition of kynurenine uptake and KYNA synthesis in the brain. To this end, the effects of proteinogenic amino acids on KYNA formation and kynurenine uptake in rat brain in vitro were comprehensively investigated [60]. Ten out of 19 amino acids (specifically, leucine, isoleucine, phenylalanine, methionine, tyrosine, alanine, cysteine, glutamine, glutamate, and aspartate) significantly reduce KYNA formation at 1 mmol/L in rat cortical slices. The amount of KYNA in the extracellular medium was reduced by 40–60% by eight amino acids (leucine, isoleucine, methionine, alanine, tyrosine, glutamine, glutamate, and aspartate), and by approximately 25% by phenylalanine and cysteine at 1 mmol/L. These amino acids show inhibitory effects in a dose-dependent manner, and partially inhibit KYNA production at physiological concentrations. Leucine, isoleucine, methionine, phenylalanine, and tyrosine, all LAT substrates, but not other five amino acids also reduce tissue kynurenine concentrations in a dose-dependent manner, and their inhibitory rates for kynurenine uptake significantly correlate with KYNA formation. IC_50_ for KYNA production and kynurenine uptake; *K*_m_ values for LATs; and physiological concentration of amino acids, including LAT substrates, are shown in Table 1. The amino acids that inhibited KYN uptake are consistent with substrate amino acids of LAT 1 rather than LAT 2, suggesting a critical role of LAT 1 in kynurenine uptake in the brain. *K*_m_ values of LAT 1 for leucine, isoleucine, methionine, phenylalanine, and tyrosine are 15–30 μmol/L, around physiological concentrations [38], indicating higher affinity than for kynurenine [39]. Furthermore, inhibition of LATs suppresses KYNA production via inhibition of kynurenine uptake in vitro and in vivo [61]. LATs inhibitor 2-aminobicyclo-(2,2,1)-heptane-2-carboxylic acid (BCH) inhibits KYNA production and kynurenine uptake in rat cortical slices in a dose-dependent manner. Administration of BCH suppresses kynurenine-induced elevations of kynurenine and KYNA levels to 50% and 70% in the mice brain. These results suggest that five LAT substrates inhibit KYNA formation via blockade of the KYN transport, while the other amino acids act via blockade of KYNA synthesis in the brain (Figure 5).

## 5. Conclusions

Recent studies have shown that KYNA modulates neurofunction by blockade of NMDA and α7nAch receptors, which is relevant to psychiatric disorders. Research has focused on pharmacologically manipulating KYNA formation to achieve the intended benefit and avoid harmful outcomes. Since humans intake several grams of amino acids from diet every day and amino acids are highly tolerable, chronic intake of amino acids may be a good tool to modulate brain function by manipulation of KYNA formation in the brain. This approach may be useful in the treatment and prevention of neurological and psychiatric diseases associated with elevated KYNA levels.

## Figures and Tables

**Figure 1 nutrients-12-01403-f001:**
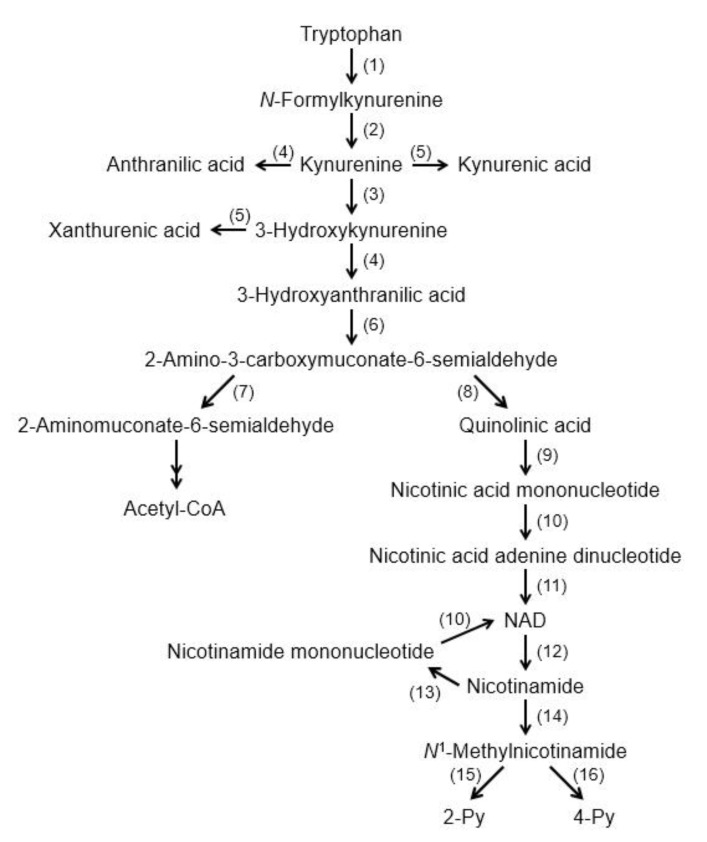
Tryptophan degradation pathway. (1) Tryptophan 2,3-dioxygenase/Indoleamine 2,3-dioxygenase (2) formamidase, (3) kynurenine 3-monoxygenase, (4) kynureninase, (5) kynurenine aminotransferase, (6) 3-hydroxyanthranilic acid oxygenase, (7) 2-amino-3-carboxymuconate-6-semialdehyde decarboxylase, (8) nonenzymatic reaction, (9) quinolinate phosphoribosyltransferase, (10) nicotinic acid (nicotinamide) mononucleotide adenylyltransferase, (11) NAD^+^ synthetase, (12) NAD^+^ degrading enzyme, (13) nicotinamide phosphoribosyltransferase, (14) nicotinamide methyltransferase, (15) 2-Py-forming *N*^1^-methylnicotinamide oxidase, and (16) 4-Py-forming *N*^1^-methylnicotinamide oxidase. Abbreviations: NAD^+^: nicotinamide adenine dinucleotide; 2 Py: *N*^1^-methyl-2-pyridone-5-carboxamide; and 4 Py: *N*^1^-methyl-4-pyridone-3-carboxamide.

**Figure 2 nutrients-12-01403-f002:**
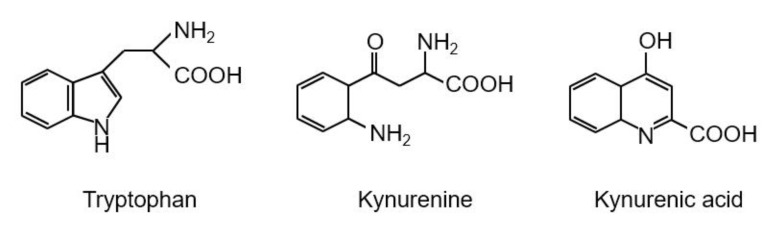
Structures of tryptophan, kynurenine, and kynurenic acid.

**Figure 3 nutrients-12-01403-f003:**
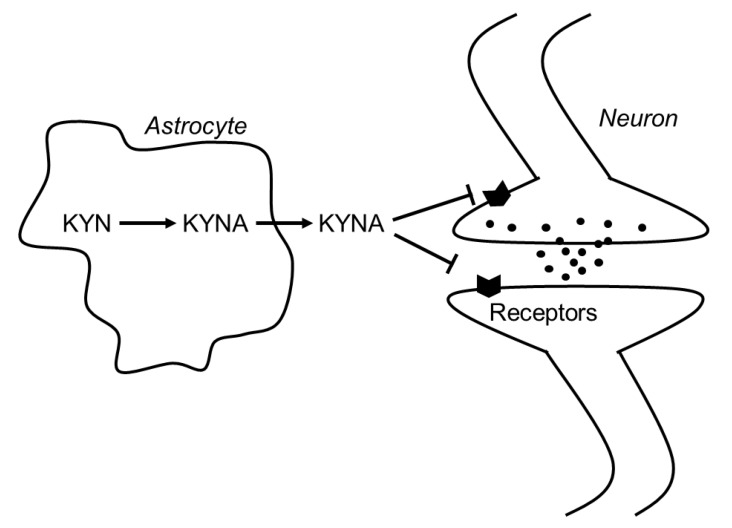
Schematic representation of the interaction between KYNA and neurotransmitters in the brain. Abbreviations: KYN: kynurenine; KYNA: kynurenic acid.

**Figure 4 nutrients-12-01403-f004:**
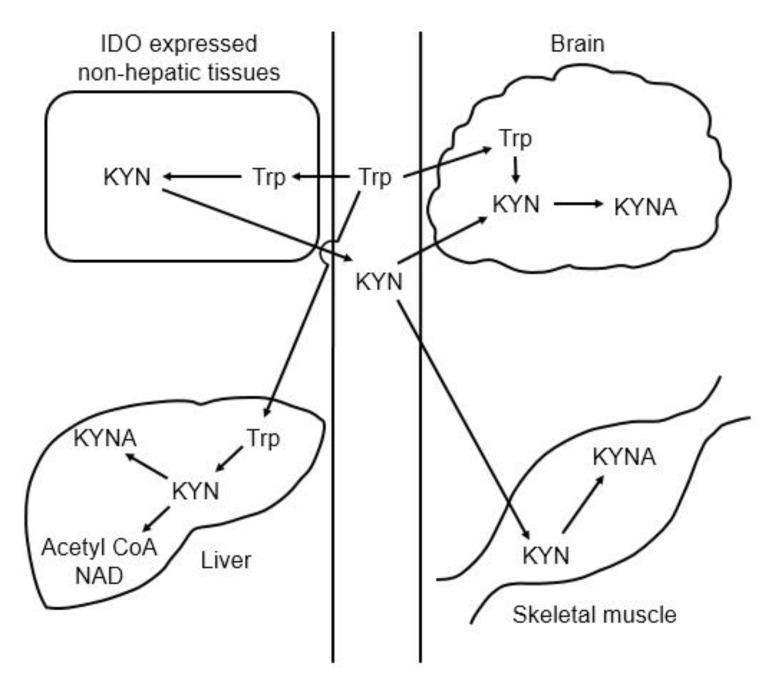
Schematic representation of the organ–organ interaction for tryptophan and kynurenine metabolism. Abbreviations: IDO: indoleamine 2,3-dioxygenase; KYN: kynurenine; KYNA: kynurenic acid; Trp: tryptophan.

**Figure 5 nutrients-12-01403-f005:**
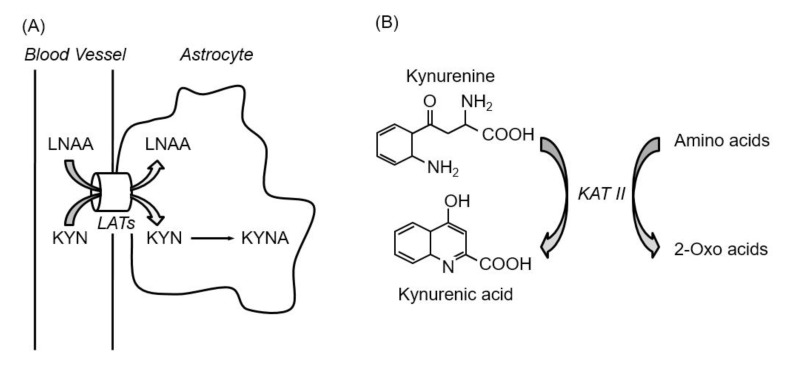
Inhibition of kynurenine uptake via LATs by amino acids (**A**), and inhibition of kynurenic acid synthesis reaction by amino acids (**B**). Abbreviations: KAT II; kynurenine aminotransferase II, KYN; kynurenine, KYNA; kynurenic acid, LATs; large neutral amino acid transporters, LNAA; large neutral amino acids.

**Table 1 nutrients-12-01403-t001:** Parameters of amino acids for KYNA production, kinetics, and physiological levels [38,39,41,60].

	IC_50_ (μmol/L)		*K*_m_ (μmol/L)		Plasma Level (μmol/L)
	KYNA Production	Kynurenine Uptake	hLAT1	rLAT2	
Leucine	36.9	30.4	19.7	119	153
Phenylalanine	22.5	10.4	14.2	45.0	58
Isoleucine	60.1	83.6	25.1	96.7	85
Methionine	184	98.6	20.2	204	54
Tyrosine	970	159	28.3	35.9	64
Histidine	–	–	12.7	181	69
Valine	–	–	47.2	–	194
Glutamate	94.9	–	–	–	77
Cysteine	110	–	–	109	11
Alanine	146	–	–	187	377
Aspartate	502	–	–	80.7	12
Glutamine	647	–	1640	151	711

IC_50_ values for KYNA production and kynurenine uptake were determined in rat cortical slices [60]. *K*_m_ values for expressed human LAT1 and rat LAT2 were determined in *Xenopus* oocytes [38,41]. Plasma free amino acids were determined in rats [39].

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
