# Peer review of "Possibility of Amino Acid Treatment to Prevent the Psychiatric Disorders via Modulation of the Production of Tryptophan Metabolite Kynurenic Acid"

_nutrients, 2020, doi:10.3390/nu12051403_

Round 1

Reviewer 1 Report

This is a relatively brief but timely review of a rapidly evolving field of research, namely the role of the major tryptophan metabolite kynurenine and its downstream metabolite kynurenic acid in brain function and dysfunction. The author is an experienced and well-regarded investigator in this area.The structure and message of the paper are therefore clear, accurate and informative.

The following changes would further improve the quality of the review:

1)In view of the title of the paper, the author should be encouraged to provide one or two Tables that summarize the original literature on the role of various amino acids as potential regulators of kynurenine neurobiology.In other words, the reader would be helped by reviewing details in an easily accessible format.This should include a listing of organ differences and of the effectiveness of the various amino acids, including kynurenine, as substrates of LAT I and LAT II.Always with appropriate references as part of the Table(s).

2)Although the author appropriately addresses the effects of amino acids on circulating kynurenine, the question if and how LATs influence the concentration of kynurenic acid in the blood deserves special attention.This is especially important since measurement of peripheral kynurenic acid is increasingly used as an indicator of brain kynurenic acid function and its impairment in humans.Reflection regarding the validity of this inference is very often missing and can lead to false interpretations and conclusions.Therefore, in a separate paragraph, the author should explain the role of amino acids and LATs in determining blood kynurenic acid levels and present arguments why (or why not) these measurements should be used as biomarkers in clinical settings.

Minor point:line 194: delete: “not from liver”

Author Response

I am grateful to Reviewer 1 for the critical comments and useful suggestions to improve my manuscript. As indicated in the responses, I revised the manuscript.

Comment 1:

In view of the title of the paper, the author should be encouraged to provide one or two Tables that summarize the original literature on the role of various amino acids as potential regulators of kynurenine neurobiology. In other words, the reader would be helped by reviewing details in an easily accessible format. This should include a listing of organ differences and of the effectiveness of the various amino acids, including kynurenine, as substrates of LAT I and LAT II. Always with appropriate references as part of the Table(s).

Response to Comment 1:

Since other reviewer suggested to describe more details of amino acids to inhibit KYNA production and kynurenine uptake in section 4, I added table 1 putting Km values to LAT 1 and 2 together with IC50 values and plasma concentrations.

Comment 2:

Although the author appropriately addresses the effects of amino acids on circulating kynurenine, the question if and how LATs influence the concentration of kynurenic acid in the blood deserves special attention. This is especially important since measurement of peripheral kynurenic acid is increasingly used as an indicator of brain kynurenic acid function and its impairment in humans. Reflection regarding the validity of this inference is very often missing and can lead to false interpretations and conclusions. Therefore, in a separate paragraph, the author should explain the role of amino acids and LATs in determining blood kynurenic acid levels and present arguments why (or why not) these measurements should be used as biomarkers in clinical settings.

Response to Comment 2:

I moved the statements about the factors to affect kynurenic acid production from section 4 to 3 corresponding to lines 261–271, and added the statements about the relationships between brain and peripheral kynurenic acid in lines 271–280.

Reviewer 2 Report

This is a concise, well-written overview of a topic which merits a wider audience, namely the role of simple dietary aspects in regulating a pathway which underlies the functioning of many organs and tissues.

There is only one major cause of concern. The paper places some emphasis on the interaction between kynurenic acid and nicotinic receptors.  This idea has been highly controversial for several years, since several attempts to repeat the original results have failed.  However, a major review has been published recently which reviews all data extensively and comprehensively, with the conclusion that overall there is no reliable or reproducible evidence for and action of kynurenic acid on nicotinic receptors.  Therefore the author should remove those statements and redraws the figures which relate to nicotinic receptors.  I would recommend inserting a reference to the new review (Stone TW 2020, Journal of Neurochemistry, 152, 627-649.)

I was also surprised that the original work on quinolinic and kynurenic acid by the same author were not included in the reference list.

Author Response

Reviewer 2

I am grateful to Reviewer 2 for the critical comments and useful suggestions to improve my manuscript. As indicated in the responses, I revised the manuscript.

Comment:

There is only one major cause of concern. The paper places some emphasis on the interaction between kynurenic acid and nicotinic receptors. This idea has been highly controversial for several years, since several attempts to repeat the original results have failed. However, a major review has been published recently which reviews all data extensively and comprehensively, with the conclusion that overall there is no reliable or reproducible evidence for and action of kynurenic acid on nicotinic receptors. Therefore the author should remove those statements and redraws the figures which relate to nicotinic receptors. I would recommend inserting a reference to the new review (Stone TW 2020, Journal of Neurochemistry, 152, 627-649.)

Response to Comment:

I removed the statements about the relationship between kynurenic acid and nicotinic receptors from section 2, and redrawn figure 3. I added the statements about the controversy over the relationship between kynurenic acid and nicotinic receptors in lines 109–113.

Reviewer 3 Report

This is a good and useful review that examines a potentially important therapeutic avenue for the treatment of psychiatric conditions.

The author mostly covers the field well - but I do feel that there are several areas where the author should provide extra details. For instance, the 2nd paragraph of sub-section 4 'Effects of amino acids on kynurenic acid production' needs more details. Specifically, the author should outline the degree of partial inhibition of KYNA production by the amino acids.

Additionally, there are many and frequent minor grammatical errors, please see attached file for edit suggestions.

Author Response

Reviewer 3

I am grateful to Reviewer 3 for the critical comments and useful suggestions to improve my manuscript. As indicated in the responses, I revised the manuscript.

Comment 1:

The author mostly covers the field well - but I do feel that there are several areas where the author should provide extra details. For instance, the 2nd paragraph of sub-section 4 'Effects of amino acids on kynurenic acid production' needs more details. Specifically, the author should outline the degree of partial inhibition of KYNA production by the amino acids.

Response to Comment 1:

I added more details of the effect of amino acids on kynurenic acid production in lines 297–345, and table 1 described parameters of amino acids such as IC50 values to kynurenic acid production and kynurenine uptake, Km values to LAT1 and 2, and plasma concentration in rats. I also added the statements about the degree of partial inhibition of KYNA production by the amino acids in lines 299–329.

Comment 2:

Additionally, there are many and frequent minor grammatical errors, please see attached file for edit suggestions.

Response to Comment 2:

I corrected the manuscript to meet reviewer 3’s suggestion.

Round 2

Reviewer 2 Report

The editors have responded to my comments and the paper is now acceptable for publication